# A time-series analysis of blood-based biomarkers within a 25-year longitudinal dolphin cohort

Aaditya V. Rangan[1]*, Caroline C. McGrouther[1], Nivedita Bhadra[2], Stephanie Venn-Watson[3], Eric D. Jensen[4], Nicholas J. Schork[2,3]

**1** Courant Institute of Mathematical Sciences, New York University, New York, New York, United States of America, **2** Quantitative Medicine and Systems Biology, The Translational Genomics Research Institute, Phoenix, Arizona, United States of America, **3** Seraphina Therapeutics, Inc., San Diego, California, United States of America, **4** US Navy Marine Mammal Program, Naval Information Warfare Center Pacific, San Diego, California, United States of America

\* avr209@nyu.edu

**Data Availability Statement:** The data is publicly available at: http://www.pnas.org/lookup/doi/10.1073/pnas.1918755117#supplementary-materials.

## Abstract

Causal interactions and correlations between clinically-relevant biomarkers are important to understand, both for informing potential medical interventions as well as predicting the likely health trajectory of any individual as they age. These interactions and correlations can be hard to establish in humans, due to the difficulties of routine sampling and controlling for individual differences (e.g., diet, socio-economic status, medication). Because bottlenose dolphins are long-lived mammals that exhibit several age-related phenomena similar to humans, we analyzed data from a well controlled 25-year longitudinal cohort of 144 dolphins. The data from this study has been reported on earlier, and consists of 44 clinically relevant biomarkers. This time-series data exhibits three starkly different influences: (A) directed interactions between biomarkers, (B) sources of biological variation that can either correlate or decorrelate different biomarkers, and (C) random observation-noise which combines measurement error and very rapid fluctuations in the dolphin's biomarkers. Importantly, the sources of biological variation (type-B) are large in magnitude, often comparable to the observation errors (type-C) and larger than the effect of the directed interactions (type-A). Attempting to recover the type-A interactions without accounting for the type-B and type-C variation can result in an abundance of false-positives and false-negatives. Using a generalized regression which fits the longitudinal data with a linear model accounting for all three influences, we demonstrate that the dolphins exhibit many significant directed interactions (type-A), as well as strong correlated variation (type-B), between several pairs of biomarkers. Moreover, many of these interactions are associated with advanced age, suggesting that these interactions can be monitored and/or targeted to predict and potentially affect aging.

**Funding:** This study is supported by the National Institute of Health and the National Institute of Aging to: N.J.S. U19 AG056169-01A1, N.J.S. U2C CA252973, N.J.S. UH3 AG064706, N.J.S. UH2 AG06470602S1, N.J.S. A.V.R. U19AG023122. The funders had no role in study design, data collection and analysis, decision to publish, or preparation of the manuscript.

**Competing interests:** I have read the journal's policy and the authors of this manuscript have the following competing interests: S.V. is a co-founder of and employed by Epitracker, Inc and Seraphina Therapeutics, Inc, which hold exclusive licensing rights from the U.S. Navy to commercialize odd-chain saturated fatty acids as human and animal health products.

## Author summary

The body is a very complicated system with many interacting components, the vast majority of which are practically impossible to measure. Furthermore, it is still not understood how many of the components that we *can* measure influence one another as the body ages. In this study we try and take a small step towards answering this question. We use longitudinal data from a carefully controlled cohort of dolphins to help us build a simple model of aging. While the longitudinal data we use does measure many important biomarkers, there are obviously a much larger number of biomarkers that haven't been measured. Our simple model accounts for these 'missing' measurements by assuming that their accumulated effect is similar to a kind of 'noise' often used in the study of complicated dynamical systems. With this simple model we are able to find evidence of several significant interactions between these biomarkers. The interactions we find may also play a role in the aging of other long-lived mammals, and may be worth investigating further to better understand human aging.

## Introduction

A better understanding of the biology of aging can help us discover strategies to stay healthier longer [1, 2]. Studying aging directly in humans has many advantages, and can help pinpoint some of the mechanisms responsible for predicting and controlling aging rates [3–7]. Unfortunately, human studies have several disadvantages, including difficulties acquiring regular samples, as well as controlling for the differences between individuals (e.g., diet, socioeconomic status and medication) [6, 8, 9]. Many of these limitations can be overcome by focusing on short-lived animals such as worms, flies and mice [10, 11], but it is important to complement this work with studies of animals that have lifespans similar to humans (i.e., with evolutionary adaptations that allow them to live 50 or more years long).

Bottlenose dolphins (*Tursiops truncatus*) provide a useful model organism, as they are long-lived mammals which perform complex social and cognitive tasks over the course of their lives [12–15]. Importantly, dolphins also share many genetic and biogerontological qualities with humans [14, 16–22]. For example, dolphins exhibit some similarities to humans regarding cellular function [23], and reduce their energy expenditure as they age [24]. Dolphins also develop several age-related conditions—such as chronic inflammation and insulin-resistance —with clinical signs similar to humans [18, 22, 25–27], while also developing histological lesions similar to certain human disorders [26–30].

In this manuscript we analyze data taken from a longitudinal cohort of 144 US Navy bottlenose-dolphins. These dolphins were cared for over three generations and were routinely sampled to asess the trajectory of 44 clinically relevant biomarkers over their lifespan, as described in [20, 31, 32]. The diet and environment of these dolphins was carefully controlled, resulting in the dolphins living an average of 32.5 years (c.f. wild dolphins live an average of 20 years) [33, 34]. The uniformity of environment, diet and health-care within this cohort implies that age-related differences between dolphins might be due to inherent or genetic factors influencing senescence [35].

This data-set was used previously to demonstrate differences in aging-rates between individual dolphins [35]. Motivated by these results, we further analyze the trajectories of each of the biomarkers for each of the dolphins within this data-set, searching for hints of any causal interactions between the measured biomarkers. In this analysis it is crucial to account for the shared fluctuations (termed 'shared variation') between biomarkers, as these shared

fluctuations typically dwarf the more subtle causal interactions we are trying to find. To account for this shared variation, we model the longitudinal data as a linear stochastic-differential-equation (SDE), extracting model parameters as indicators of which biomarkers might affect one another. After doing so, we are able to identify interactions that are associated with aging. These interactions may play a role in the biology of aging, and are prime candidates for further investigation.

## Results

### Motivation for the model

Obviously, the body is a very complicated dynamical system [36]. Any set of biomarkers can only ever tell part of the story, and there will always be an abundance of unmeasured factors that play an important role in the trajectory of all the measured biomarkers. Given any particular subset of (observed) biomarkers, we typically expect the changes in those biomarkers to come from three different categories.

**type-A: Directed interactions**. Any particular biomarker may play a direct role in influencing the dynamics of other biomarkers. For example, one biomarker might 'excite' or 'inhibit' another, with an increase in the first promoting an increase (or, respectively, decrease) in the second.

**type-B: Shared biological variation**. As the system evolves, we expect a large number of unmeasured biomarkers to affect each of the measured biomarkers. Generally, this multitude of unmeasured effects will accumulate, giving rise to a combined effect that is essentially unpredictable given the measurements at hand. Collectively, the influence of these unmeasured effects can seem random.

**type-C: Observation-noise**. To complicate things further, our measurements of each observed biomarker may not be perfectly accurate. Typically there are additional sources of 'observation-noise' which introduce random variation to our measurements. This type-C variation can include both measurement errors as well as true biological fluctuations that are much more rapid than the typical time-interval of the experimental measurements themselves (e.g,. hourly variations based on the individual's activity).

To understand how large a role each category plays, we can look at the changes in each of the biomarkers within the dolphin data-set. One example for the biomarker 'Alkaline Phosphatase' (AlkPhos) is shown in Fig 1. This figure illustrates the distribution of measured variable-increments (between one time-point and the next) for this biomarker. The large variation in these increments indicates that the type-B and type-C effects play a large role. Furthermore, there is a striking linear relationship between (i) the magnitude of the measured time-increment and (ii) the variance in the measured variable-increments. This linear relationship indicates that a significant component of the type-B variation is similar to the Brownian processes commonly seen in high-dimensional dynamical systems [37]. Such a Brownian process—also called a 'stochastic drive'—can provide a source of shared variation which correlates different biomarkers. Finally, we remark that the variation in variable-increments is still significant even when the measured time-increment is 0. In other words, multiple repeated measurements of the same dolphin may not always return the same value. This indicates that there is a significant source of type-C variation in our measurements.

The qualitative aspects of Fig 1 are typical for this dataset; Figs 2 and 3 illustrate the distribution of variable-increments for two other measured biomarkers. Note that, once again, we see that the variation in variable-increments is large, and that there is evidence for both type-B

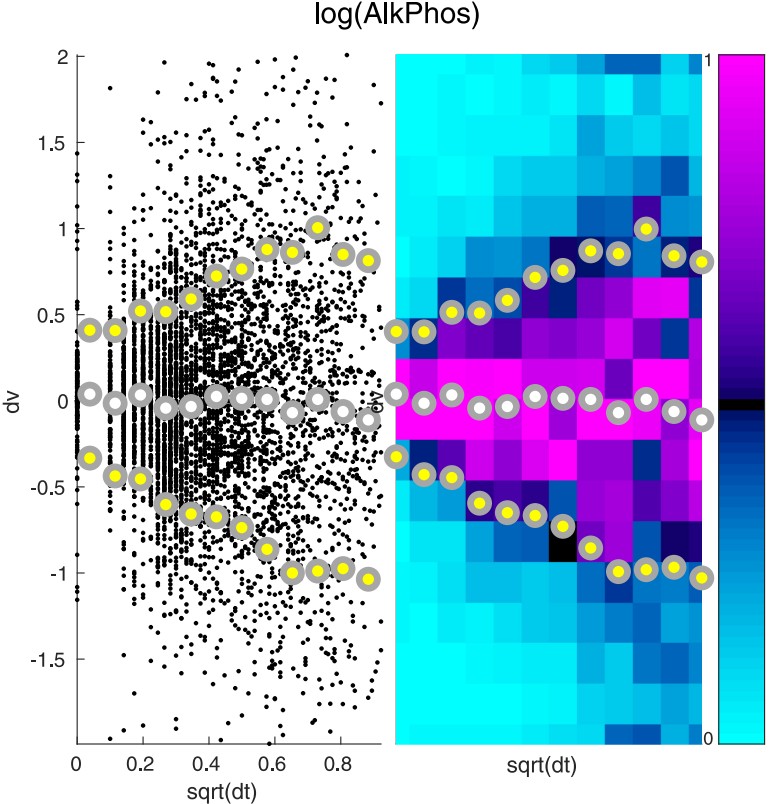

**Fig 1. Increments of Alkaline Phosphatase.** Here we illustrate the distribution of increments associated with Alkaline Phosphatase (AlkPhos). On the left we show a scatterplot of these increments. Each small black point in the background represents a pair of time-adjacent measurements in a single dolphin, with the square-root of the time-increment (in years) along the horizontal, and the change in AlkPhos along the vertical. The time-increments are then divided into 12 bins, and the mean (white dot) plus and minus one standard-deviation (yellow dots) are displayed in the foreground. On the right we show the same data, with the 12 time-bins shown as vertical strips. Each vertical strip is further divided into 18 boxes, which are colored by the number of increments within that time-bin which fall into that box (i.e., a histogram for each time-bin, scaled to have maximum 1, colored as shown in the colorbar to the far right). Note that the distribution of increments for each time-bin becomes wider as the time-increment increases. Importantly, the standard-deviation increases roughly linearly with the square-root of the time-increment (i.e., the variance increases roughly linearly with the time-increment), as expected from a Brownian-process. Note also that the variance is nonzero even when the time-increment is 0, implying that there must be an extra source of variation in addition to this Brownian-process.

and type-C variation. For some biomarkers we expect the type-B variation to be larger than the type-C variation, whereas for other biomarkers we expect the type-C variation to dominate (e.g., compare Figs 2 and 3).

In the next section we'll introduce our model for the dynamics. Within this model we'll model the type-A directed interactions as simple linear interactions (see Eq (1) below). Motivated by the observations above, our model will also include terms to account for type-B and type-C variations. we'll model the type-B variations as an (anisotropic) Brownian-process (see Eq (1) below), and we'll model the observation-noise as an (anisotropic) Gaussian with time-independent variance [38] (see Eq (2) below).

## Model structure

Given the motivation above, we can model the evolution of any $d$ specific variables over the time-interval $[t, t']$ using a simple linear stochastic-differential-equation (SDE) of the following

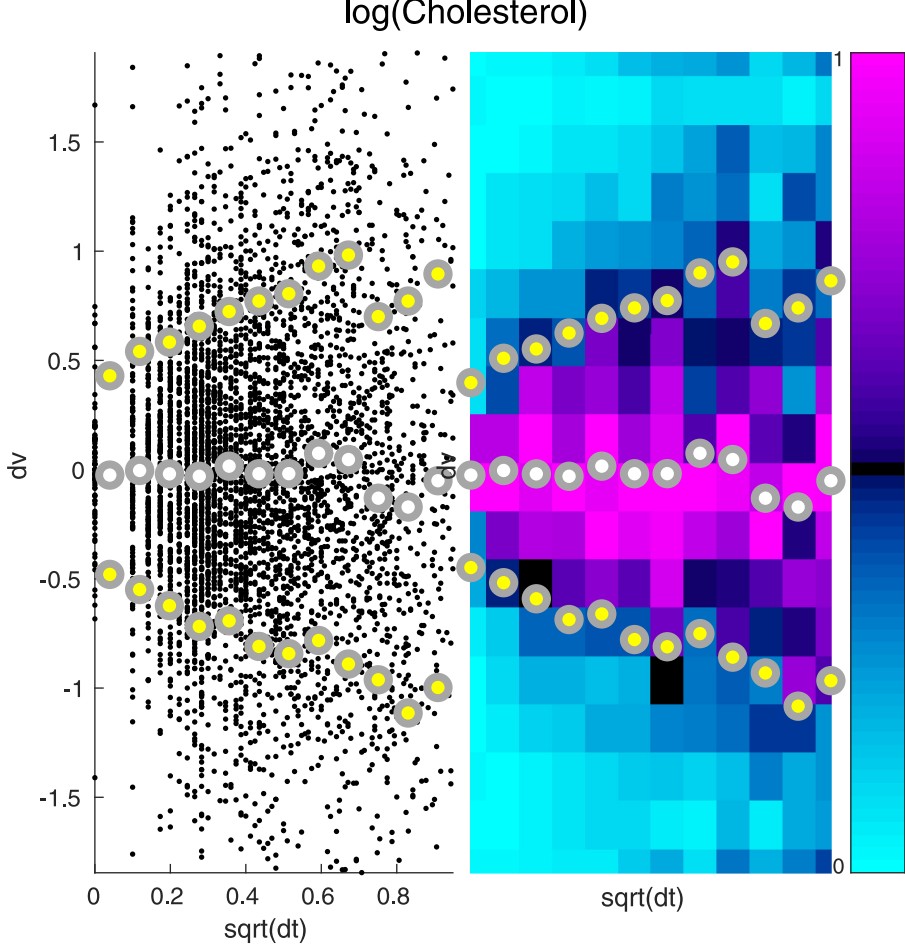

**Fig 2. Increments of Cholesterol.** Here we illustrate the distribution of increments associated with Cholesterol. The format for this figure is the same as Fig 1.

form:

$$dX(t) = [\boldsymbol{a} + A \cdot X(t)]dt + B \cdot dW(t), \tag{1}$$

where $X(t) \in \mathbb{R}^d$ represents the $d$-dimensional vector-valued solution-trajectory at the initial time $t$, the time-increment $dt = t' - t$ represents the difference between the initial time $t$ and the final time $t'$, and $dX(t) = X(t') - X(t) \in \mathbb{R}^d$ represents the vector of variable-increments between times $t$ and $t'$. We further assume that we do not measure $X(t)$ directly, but rather some $Y(t)$ which depends on $X(t)$ and which also incorporates the type-C observation-noise:

$$Y(t) = X(t) + C \cdot \boldsymbol{\epsilon}(t). \tag{2}$$

In Eqs (1) and (2) there are several terms on the right hand side which contribute to the observed dynamics.

**Baseline velocity**: The vector $\boldsymbol{a} \in \mathbb{R}^d$ corresponds to a constant 'velocity' for each of the variables. This represents the rates at which each variable would increase, were it not for the other influences within the system.

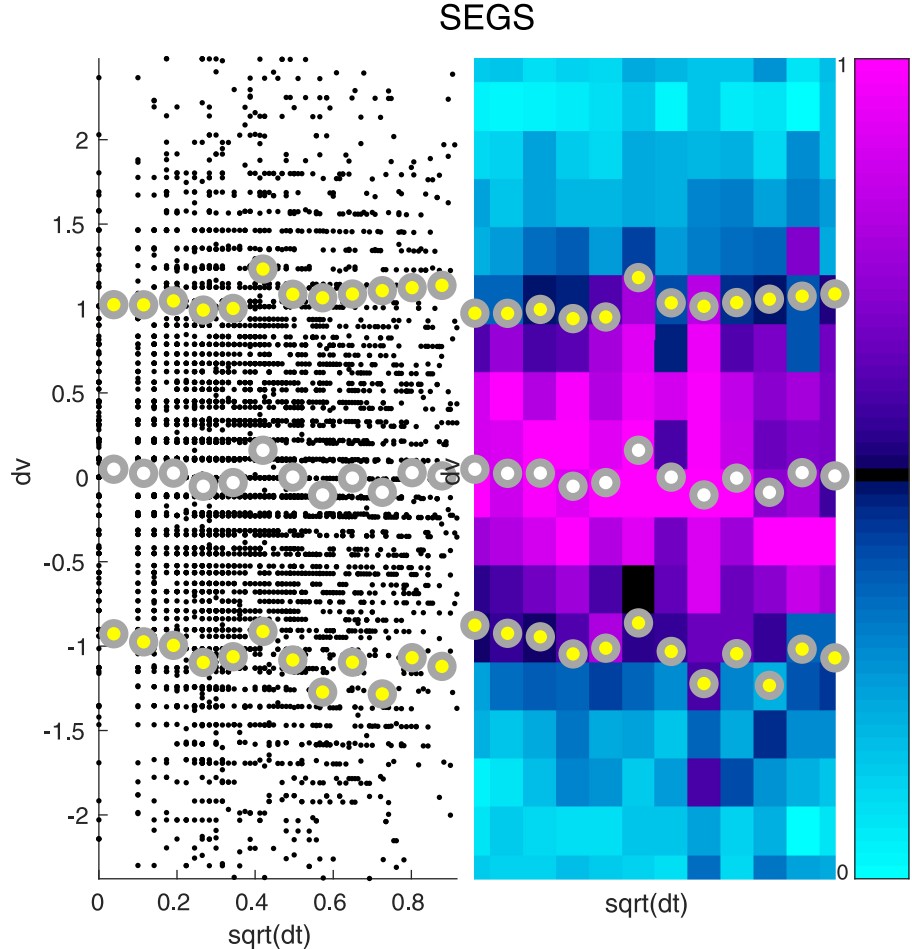

**Fig 3. Increments of Segmented Neutrophils.** Here we illustrate the distribution of increments associated with Segmented Neutrophils (SEGS). The format for this figure is the same as Fig 1.

**Directed interactions**: The matrix $A \in \mathbb{R}^{d \times d}$ represents the type-A interactions between variables. These directed interactions are modeled as linear effects; the value of a 'source' variable $v'$ will influence the rate at which the 'target' variable $v$ increases at time $t$ via the factor $A_{vv'}X_{v'}(t)$. Each of these type-A interactions can be thought of as part of a pathway or larger network describing how the variables influence one another. These directed interactions are often called 'deterministic interactions', as they do not explicitly involve any randomness.

**Stochastic drive**: The matrix $B$ represents the biological variation (due to an accumulation of independent unmeasured effects) expected to affect the model variables as they evolve. The source of the type-B variation is modeled by the noise $dW(t)$ (i.e., Brownian-increments) which are each drawn independently from the Gaussian distribution $\mathcal{N}(0, dt \cdot I_{d \times d})$. The matrix $B \in \mathbb{R}^{d \times d}$ controls the correlations between the noise received by the different variables [37]. Put another way, $dtBB^{\mathsf{T}}$ is the covariance matrix of biological variation for this model, analogous to the variance-covariance matrix in quantitative genetics [39]. This biological variation is often called a 'stochastic drive' to disambiguate it from the type-C observation-noise mentioned below.

**Observation-noise**: The matrix $C$ in Eq (2) represents both the noise inherent to the observations, as well as the effects of very rapid unpredictable fluctuations in the biomarkers. Each vector $\boldsymbol{\epsilon}(t) \in \mathbb{R}^d$ at each observation-time is drawn (independently) from the Gaussian distribution $\mathcal{N}(0, I_{d \times d})$. The matrix $C \in \mathbb{R}^{d \times d}$ controls the level of correlations within the type-C errors; the covariance of this observation-error is given by the symmetric-matrix $CC^{\mathsf{T}}$.

Note that the observation-times are not necessarily unique: $t'$ could very well equal $t$, and $dt$ may equal 0. In this situation $dX(t) \equiv \mathbf{0}$, and so $X(t')$ will equal $X(t)$. However, because of the type-C errors, $Y(t')$ will in general be different from $Y(t)$.

While we could, in principle, fit an SDE to all the variables simultaneously (with a large $d$), we opt to consider only pairwise combinations of variables (with $d = 2$ in each case). Mathematically, each such pairwise fit (for variables $v$ and $v'$) corresponds to approximating the system's trajectory after projecting the full system onto the subspace spanned by those two variables. We make this decision for three reasons.

**Increase Power**: First and foremost, the number of model parameters, as well as the number of observations required to accurately estimate these parameters, both increase as $d$ increases. We will have the most statistical power when $d$ is low. This topic is discussed in more detail within S1 Text.

**Avoid Redundancy**: Second, redundancies in any subset of variables (e.g., GFR and Creatinine, or MCH and MCV) can easily create the illusion of large interactions between the redundant set and other variables. These spurious interactions will be statistically, but not biologically significant. In mathematical terms, these redundancies will result in a poorly conditioned inverse problem, which we want to avoid [40].

**Easily Understand**: By considering only pairwise interactions, the results can be interpreted more easily. The interactions between any pair of variables can then be considered (and used to make predictions) even in situations where the other variables aren't accessible, or haven't been measured.

We acknowledge that there are many strategies for modeling the interactions in this longitudinal data-set (see, e.g., [41–45]). For example, there are many similarities between our model and vector-autoregression [46]. Indeed, if the time-intervals $dt$ in this data-set were all of equal size, then our method would be equivalent to one-step vector-autoregression; the only structural difference being that in such a scenario the type-B and type-C variation would be indistinguishable from one another. We opt for the SDE proposed above because (i) it has relatively few parameters, (ii) it is easy to understand, and (iii) it can easily accommodate the irregular time-intervals and occasional missing elements within this data-set.

## Model interpretation

We use a generalized regression (see S1 Text) to fit the longitudinal data for each pair of biomarkers $v$ and $v'$ with the simple dynamical system (SDE) above. After fitting the model parameters we interpret $A$ and $BB^{\mathsf{T}}$ as having potential biological significance.

The entries $A_{vv'}$ indicate the directed interactions (i.e., the deterministic effects) from biomarker $v'$ to biomarker $v$. If $A_{vv'}$ is positive, then we expect an increase in biomarker $v'$ to typically precede a subsequent increase in biomarker $v$; we conclude that $v'$ 'excites' $v$. Conversely, if $A_{vv'}$ is negative, then we expect that $v'$ will 'inhibit' $v$; an increase in $v'$ will typically precede a subsequent decrease in $v$.

The symmetric entries $[BB^{\mathsf{T}}]_{vv'} = [BB^{\mathsf{T}}]_{v'v}$ indicate the shared biological variation (i.e., the stochastic drive) expected to influence both biomarkers $v$ and $v'$. If $[BB^{\mathsf{T}}]_{vv'}$ is positive, then $v$

and $\nu'$ will experience correlated stochastic input which will give rise to correlated (but not causal) fluctuations between these two biomarkers. Conversely, if $[BB^\mathsf{T}]_{\nu\nu'}$ is negative, then $\nu$ and $\nu'$ will experience anti-correlated stochastic input which will produce anti-correlated (but again, not causally-linked) fluctuations between the two biomarkers. Because we fit the SDE to each pair of biomarkers individually, the correlations (or anti-correlations) within this stochastic drive to the pair $\nu$, $\nu'$ can result from the accumulated effects of the other measured biomarkers (other than $\nu$, $\nu'$), as well as from biomarkers which weren't included in the study at all.

To illustrate these kinds of relationships, we fit the SDE in Eqs (1) and (2) to the two biomarkers 'Iron' and 'Mean corpuscular hemoglobin' (MCH) across all observation-times across all dolphins. Referring to Iron and MCH as $\nu'$ and $\nu$, respectively, we find that $A_{\nu\nu'}$ is significantly positive and $A_{\nu'\nu}$ is significantly negative (with uncorrected $p$-values $p_0 < 1e - 11$ and $p_0 < 1e - 4$ respectively, see Methods). Moroever, we find that $[BB^\mathsf{T}]_{\nu\nu'}$ is significantly positive ($p_0 < 1e - 50$). As we'll discuss below, the type-B correlations are quite a bit larger (in magnitude) than the type-A interactions for these two biomarkers (note the difference in $p$-values), and an accurate estimate of $A_{\nu\nu'}$ can only be obtained after estimating $[BB^\mathsf{T}]_{\nu\nu'}$.

The directed interactions $A$ can be visualized using the phase-diagram of the linear differential-equation associated with $A$ (i.e., by ignoring the effect of $B$ in Eq (1)). This phase-diagram is shown on the left of Fig 4, with the directed interactions giving rise to the flow vectors in the background (grey). For this differential-equation a surplus of Iron will drive an increase in

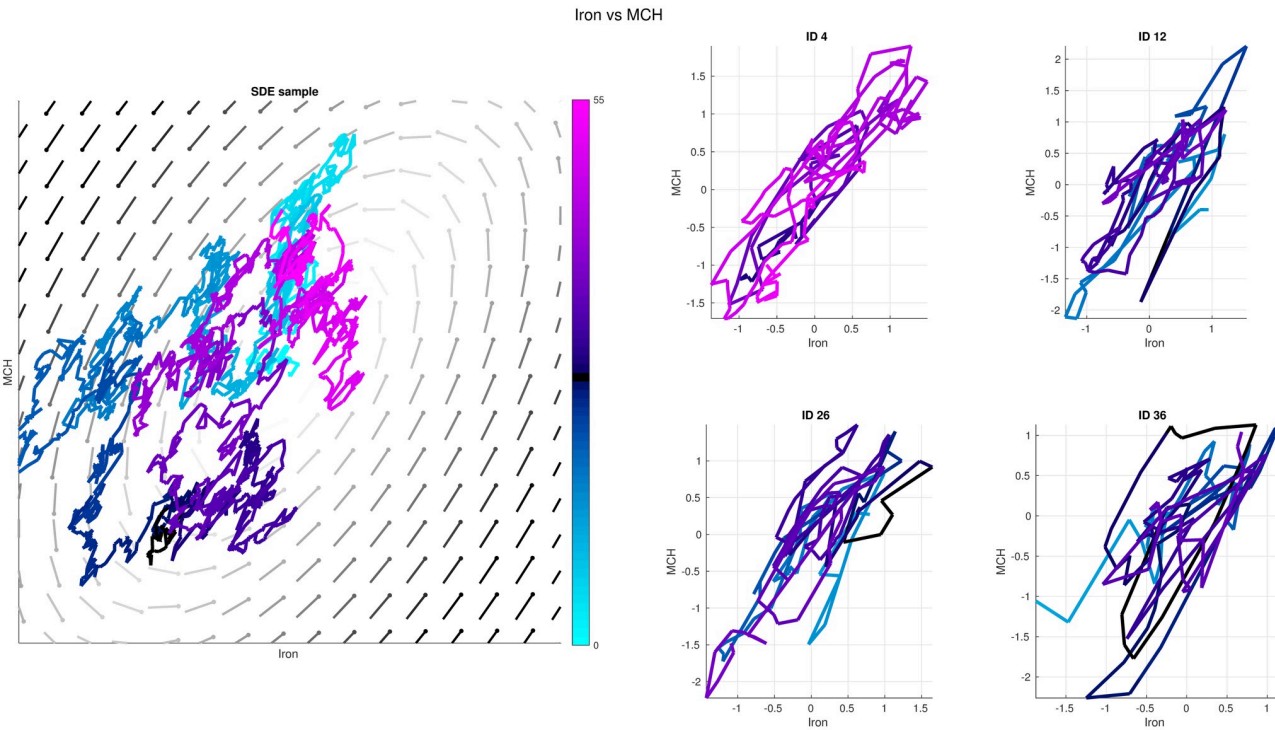

**Fig 4. Model of Iron vs MCH: A.** Here we illustrate the directed interactions between Iron (horizontal) and mean corpuscular hemoglobin (MCH) (vertical). The left-hand-side shows the phase-diagram associated with $A$, as well as a sample-trajectory from the SDE (including $B$). Other sample trajectories are shown in Fig J in S1 Text. The right-hand-side shows several subplots from various dolphins (after kalman filtering to reduce the observation error). The colorscale refers to age (see inset colorbar).

MCH, while a surplus of MCH will drive a decrease in Iron. This combination of excitation and inhibition will tend to give rise to 'counterclockwise' motion within the Iron-MCH phase-plane (indicated by the directions of the grey arrowheads).

The subplot on the left of Fig 4 also shows a finely sampled trajectory $X(t)$ of the SDE (which includes the effect of the nonzero $B$). This trajectory is colored by age, from 0yr (cyan) to 55yr (magenta) (see adjacent colorbar). Note that the sample-trajectory often deviates from the counterclockwise course suggested by the directed interactions, often meandering around due to the stochastic drive provided by $B$. The sample trajectory shown here is just one possible trajectory for the SDE; other sample trajectories are shown in Fig J in S1 Text. Subplots illustrating the (kalman filtered) trajectories of several individual dolphins are shown on the right of this figure, using the same colorscale. Similar to the sample trajectory from the SDE, these experimental trajectories also meander around, while exhibiting a global tendency to prefer counterclockwise loops over clockwise ones.

The type-B and type-C variations in these two biomarkers are shown in Fig 5. To illustrate these sources of variation, we take all the time-increments $dt$ used to fit the SDE (across all pairs of adjacent observation-times and all dolphins), and divide them into 10 percentile-bands. For each percentile-band of $dt$ we construct a scatterplot of the observed variable-increments $dY(t) \in \mathbb{R}^2$ for these two biomarkers. We fit the distribution of variable-increments in each percentile-band with a Gaussian, and indicate the 1- and 2-standard-deviation contours of this Gaussian with bright- and pale-yellow ellipses, respectively. The covariance-matrix $CC^{\mathsf{T}}$ is approximately equal to the covariance of the scatterplot shown in the upper-left, with

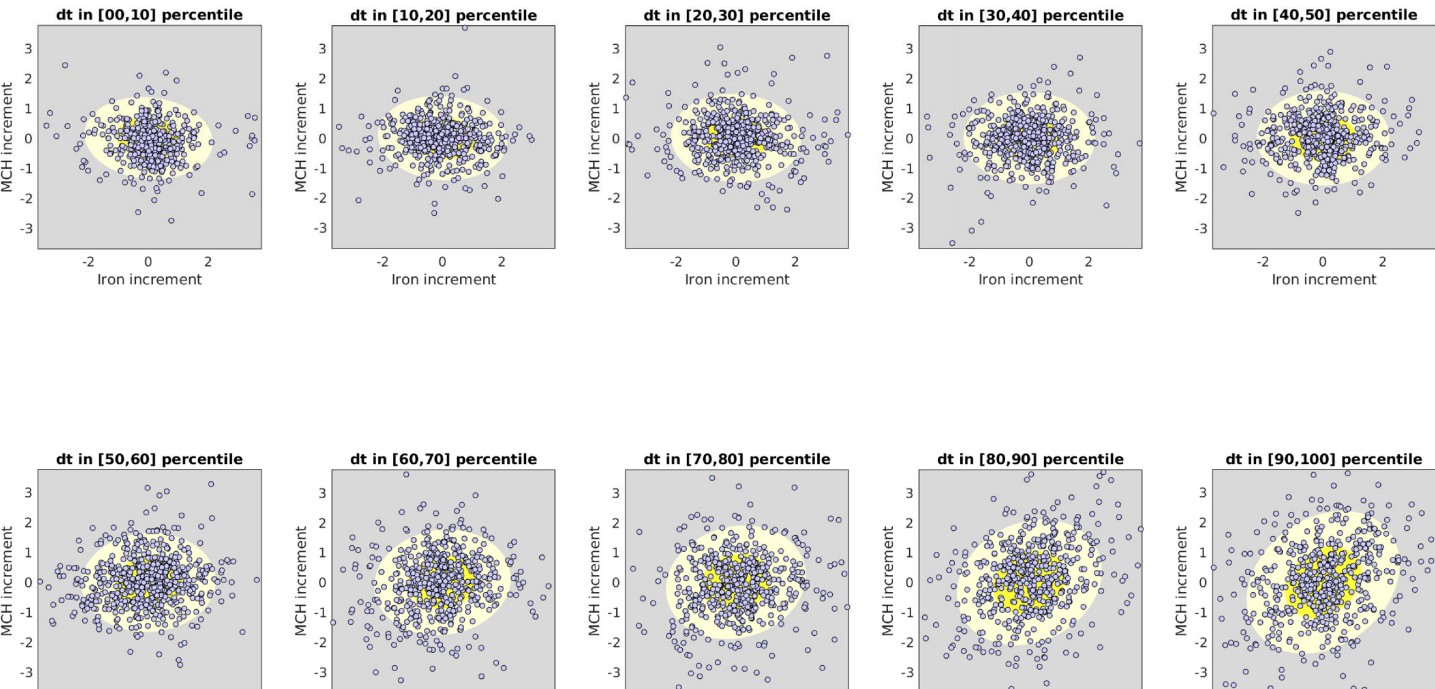

**Fig 5. Model of Iron vs MCH: B and C.** Here we illustrate the type-B and type-C variations associated with Iron (horizontal) and MCH (vertical). Each subplot shows a scatterplot of variable-increments for a percentile-band of observed time-increments. The yellow ellipses correspond to standard-deviation contours of the best-fit Gaussian distribution within each scatterplot.

$dt \sim 0$. The covariance-matrix for each of the other subplots is roughly given by $dtBB^{\mathrm{T}} + CC^{\mathrm{T}}$; as $dt$ increases the variable-increments become more and more correlated, indicating that $[BB^{\mathrm{T}}]_{vv'}$ is positive (i.e., these two biomarkers are influenced by a shared and correlated stochastic drive).

For these two biomarkers the type-B and type-C variations contribute heavily, dominating the evolution of the trajectory of the SDE. Specifically, the term $B$ in Eq (1) is several times larger than the matrix $A$. The term $C$ in Eq (2) is also large, contributing observation-noise which is a few times larger (on average) than the type-A interactions. This is quite common, and most biomarker-pairs exhibit a similar phenomenon.

If we were to ignore the type-B and/or type-C interactions when fitting this data (i.e., setting $B \equiv 0$ in Eq (1) or $C \equiv 0$ in Eq (2)), then the correlations in these variations would impact our recovery of $A$, giving rise to inaccurate estimates of $A_{vv'}$. These spurious estimates for $A_{vv'}$ would, in turn, give rise to an abundance of false-positives and false-negatives when estimating the significance of the directed interactions across the biomarker-pairs within the data-set. By accounting for these type-B and type-C terms within the SDE, we believe that we can accurately estimate the size and direction of the type-A interactions.

To summarize, we expect Iron and MCH to exhibit strong correlated (but not necessarily causal) fluctuations due to shared input coming from the accumulated effect of other (potentially unmeasured) biomarkers. By taking into account these strong type-B correlations, we can attempt to tease out more subtle directed type-A interactions. For these two biomarkers we do indeed find evidence of a statistically significant directed link: an increase in Iron will tend to produce a subsequent increase in MCH, while an increase in MCH will tend to produce a subsequent reduction in Iron.

## Longitudinal analysis

By fitting the longitudinal data with the SDE Eqs (1) and (2) above, we observe multiple significant (type-A) directed interactions, as well as (type-B) shared biological variation. An illustration of the type-A interactions observed across all dolphins and ages is shown in Fig 6, while the corresponding type-B correlations are shown in Fig 7.

We first remark that the type-B correlations typically dominate the type-A interactions. We saw an example of this earlier when considering the case of Iron versus MCH, and the same general trend holds for most of the biomarker pairs. Many strong type-B correlations can be observed between the biomarkers relating to red blood cells (RBCs) and hemoglobin (HGB), such as Iron, RBC, HGB, HCT, MCV, MCH and MCHC. These correlations are to be expected, because many of these biomarkers are influenced by the same underlying biological factors [47, 48]. Similarly, we see many strong type-B correlations between the biomarkers relating to liver function, such as LDH, AST, ALT and GGT. Once again, these are all influenced by the same factors, and strong correlations are to be expected [49, 50].

Regarding the type-A interactions, we note that most of the more significant type-A interactions involve an $A_{vv'}$ and $A_{v'v}$ with opposite signs. That is, one biomarker excites the other, while being inhibited in return (see, e.g., the interaction between Iron and MCH mentioned earlier). These 'push-pull' pairs manifest when the observed data for biomarkers $v$ and $v'$ both look like noisy versions of the same rough trajectory (e.g., meandering back-and-forth), but with one of the biomarkers 'leading' the other in time. The linear SDE that best fits this scenario is usually one which gives rise to stochastic oscillations with a distribution of temporal correlations fit to what was observed in the real data.

Some of the more statistically significant push-pull interactions include: (i) Iron $\rightleftharpoons$ MCH, (ii) HGB $\rightleftharpoons$ SED60, (iii) HCT+MCH $\rightleftharpoons$ Bilirubin, and (iv) HGB $\rightleftharpoons$ MCH. These interactions

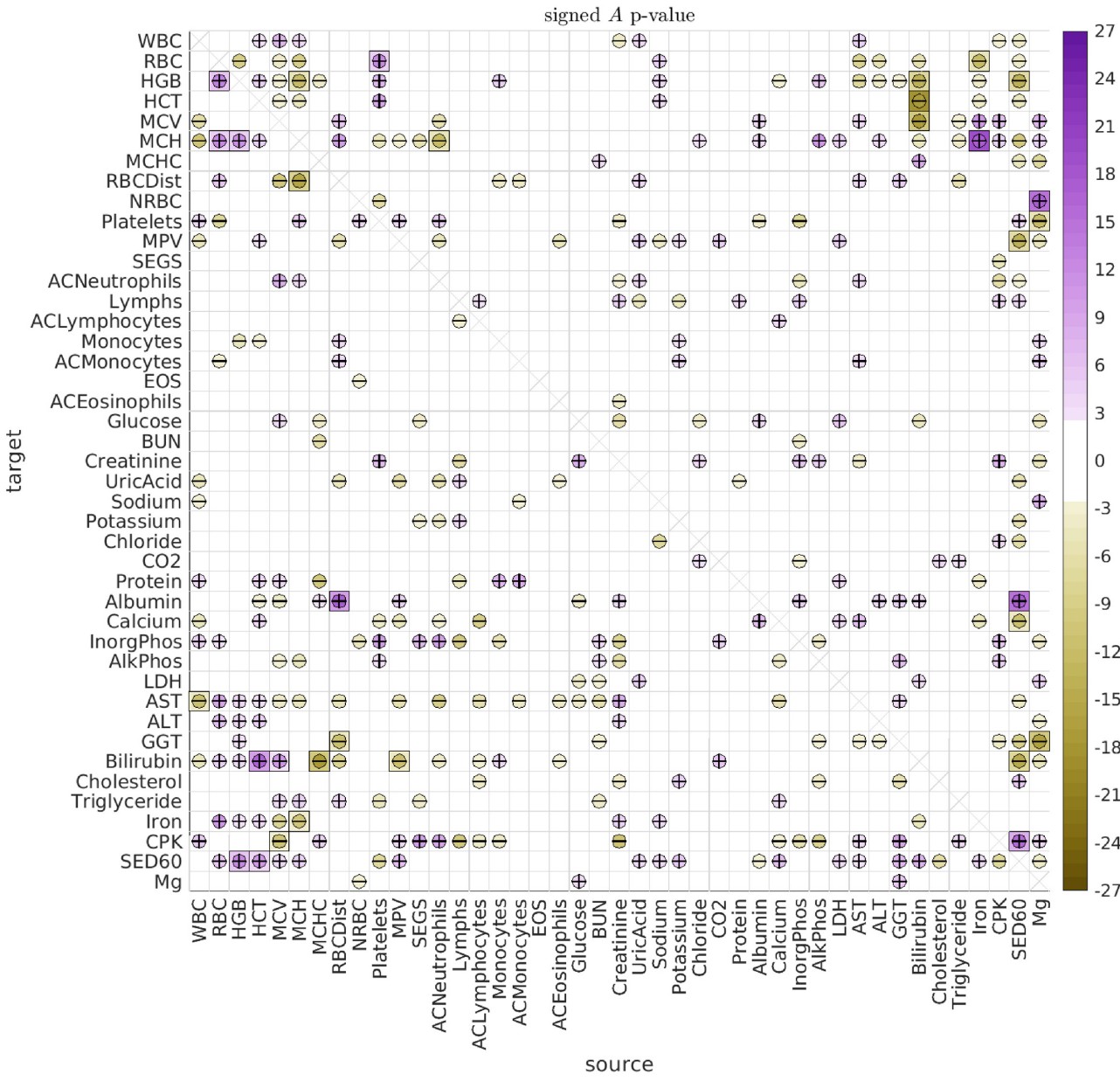

**Fig 6. Significant A.** Here we illustrate the significant directed interactions $A_{vv'}$ observed when pooling all dolphins and all ages. The 'source' biomarker $v'$ is shown along the horizontal, with the 'target' $v$ shown along the vertical. The color of each interaction indicates the log-$p$-value for that interaction, ascribed a sign according to the sign of the interaction. Thus, interactions are colored lavender if the interaction is positive, and goldenrod if the interaction is negative (see colorbar on the right). For ease of interpretation, interactions are also given a '+' or '-' symbol to indicate their sign. The bonferroni-corrected $p$-values $p_b$ are shown as squares, while the inscribed circles indicate the associated holm-bonferroni adjusted $p_h$.

suggest an intricate interplay between Iron, various RBC attributes, and inflammation (for which SED60 is a proxy). These interactions are not without precedent in humans. For example, inflammation is well known to have wide-ranging effects, and can alter red blood cell clearance as well as red blood cell counts [51, 52]. The production and regulation of Heme and Hemoglobin also involves many mechanisms, and can be affected by inflammation, in turn affecting endothelial cell activation [53–57]. The breakdown and/or destruction of RBCs can

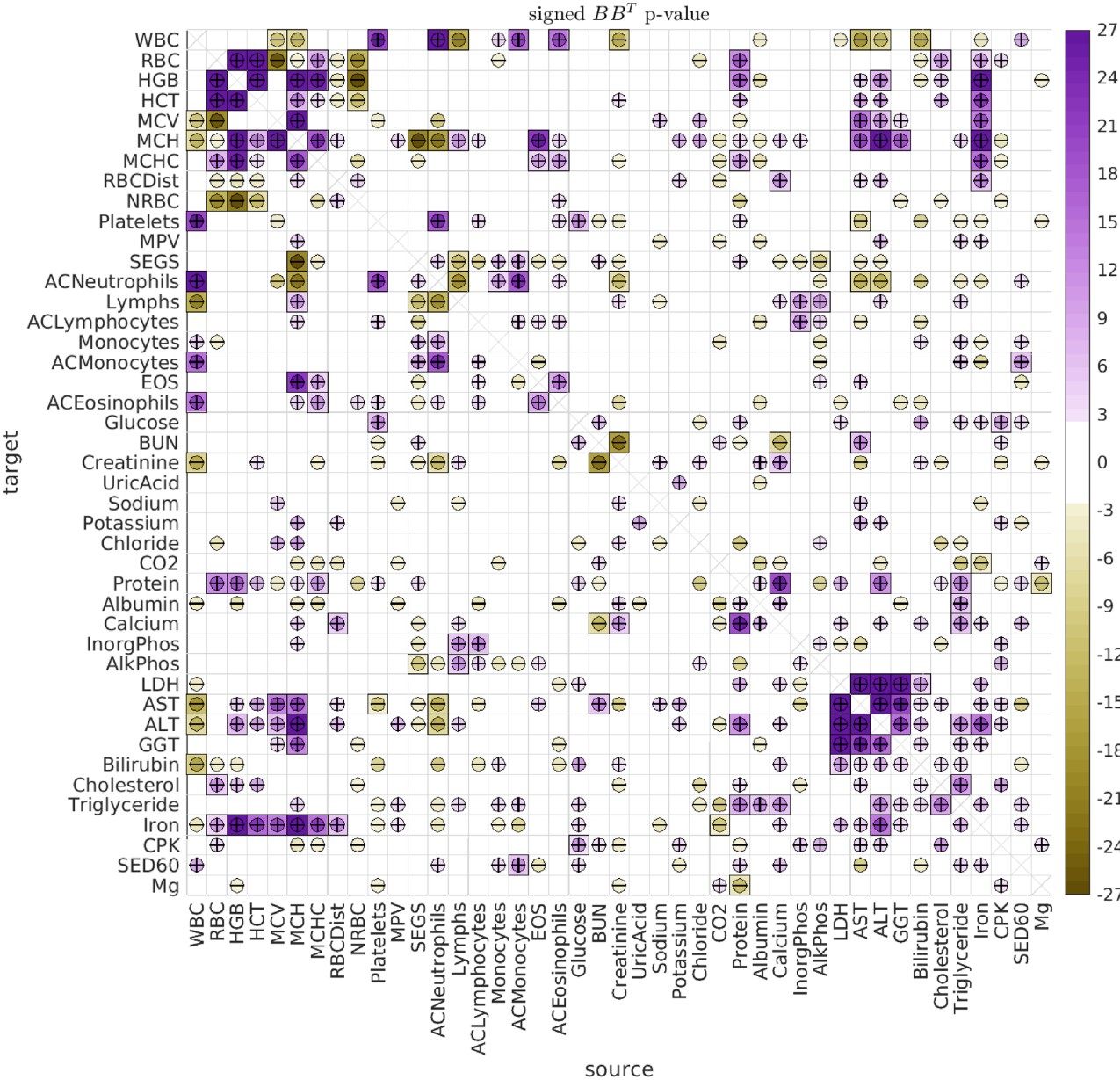

**Fig 7. Significant B.** This figure has the same format as Fig 6, showing the significance of the covariance $[BB^T]_{vv'}$. As in Fig 6, these results refer to pooling all dolphins and all ages.

also release damage-associated molecular patterns which can activate inflammatory pathways [58], and result in increased bilirubin levels, which in turn can trigger erythrocyte death [59].

**Grouping the interactions.** When observing the array of type-A interactions in Fig 6, it is clear that there are an abundance of 'push-pull' biomarker-pairs (such as Iron and MCH). Further inspection reveals that these push-pull pairs are not haphazardly scattered throughout the array (i.e., they are not 'randomly distributed'). Instead, there are subsets of biomarkers which exhibit collective push-pull interactions.

For example, let's define the subsets $\mathcal{V}$ and $\mathcal{V}'$ as follows. Let $\mathcal{V}$ contain the biomarkers RBC, HGB, and HCT, and let $\mathcal{V}'$ contain the biomarkers AST, MCH, Bilirubin, ALT, Sed60 and

Iron. With this definition, we see that these two subsets $\mathcal{V}$ and $\mathcal{V}'$ form a 'block' of push-pull interactions. That is, the directed interaction between any source biomarker from set $\mathcal{V}$ and any target biomarker from set $\mathcal{V}'$ is excitatory, while the reciprocal interaction is inhibitory. These directed interactions may be related to the observed relationships between liver function and RBC elimination [60–62].

The particular block of push-pull interactions described above is quite large (comprising a set of 3 and a set of 6 biomarkers), and is substantially larger than the typical push-pull blocks one would expect if the A-type interactions were randomly reorganized (p-value $< 1e - 21$). There are other (less significant) push-pull blocks within the array of A-type interactions as well. To systematically search for these push-pull blocks, we use a modified version of the biclustering techniques described in [63] (see S1 Text).

We can use this strategy to rearrange the biomarkers to reveal the significant push-pull blocks. As described in S1 Text, many of the significant push-pull blocks involve overlapping subsets of biomarkers, and there is no single best strategy for presenting them all. One possible rearrangement, which isolates some of the larger disjoint push-pull blocks, is shown in Fig 8. In addition to the push-pull block described above (shown in the upper left of Fig 8), there are two other push-pull blocks that can be seen towards the center of Fig 8. The first is defined by $\mathcal{V}$ containing AlkPhos and InorgPhos, and $\mathcal{V}'$ containing CPK, Platelets and BUN (p-value $< 0.007$). The second is defined by $\mathcal{V}$ containing Creatinine and Lymphs, and $\mathcal{V}'$ containing AlkPhos, InorgPhos, CPK and Platelets (p-value $< 0.026$).

When considering the type-B interactions the situation is simpler. The array of B-type interactions is symmetric, and is already quite structured. A simple spectral clustering (i.e., sorting the leading term in an eigendecomposition of $BB^{\mathsf{T}}$) can be used to reveal many groups of correlated biomarkers. In Fig 9 we use this strategy to rearrange the biomarkers to reveal several of the most significant (and disjoint) groups.

In this figure one can immediately see several different subsets of biomarkers which exhibit high levels of intra-group correlation. For example, the group of biomarkers in the bottom right corner of Fig 9 includes RBC, HCT, MCHC, HGB, Iron, MCH and MCV, all related to red blood cell function, as well as AST and ALT, proxies for stress to the liver. This group is anticorrelated with the group of biomarkers in the top left of Fig 9, including ACNeutrophils, WBC, SEGS, Platelets, ACMonocytes and Monocytes, all relating to other cell types.

Other groups can also be seen, such as the group near the center of Fig 9, including AlkPhos, InorgPhos, Lymphs, and ACLymphocytes, relating the phosphate and phosphatase concentrations to lymphocyte count.

**Age-associated interactions.** To search for age-associated interactions, we checked if there were significant differences between the older dolphins (older than 30) and the middle-aged dolphins (between 10 and 30 years old). The results for the type-A interactions are shown in Fig 10 (with the type-B correlations shown in Fig I of S1 Text). While we saw very few significant differences in the directed interactions between the male- and female-dolphins overall in this data-set (the only exceptions being the influence of magnesium on platelets and inorganic phosphate), several previous studies have found significant differences between the sexes [26, 33, 34, 64]. Consequently, we repeated our analysis after segregating by sex. These results are shown in Figs 11 and 12.

While our sex-segregated analysis has lower power than when the sexes were pooled, we can already see that many of the significant differences observed in the general population are in fact driven by one sex or the other. For example, the male dolphin population drives the age-related change in the push-pull directed interaction between RDW and MCV. On the other hand, the female dolphin population drives many of the other observed age-related

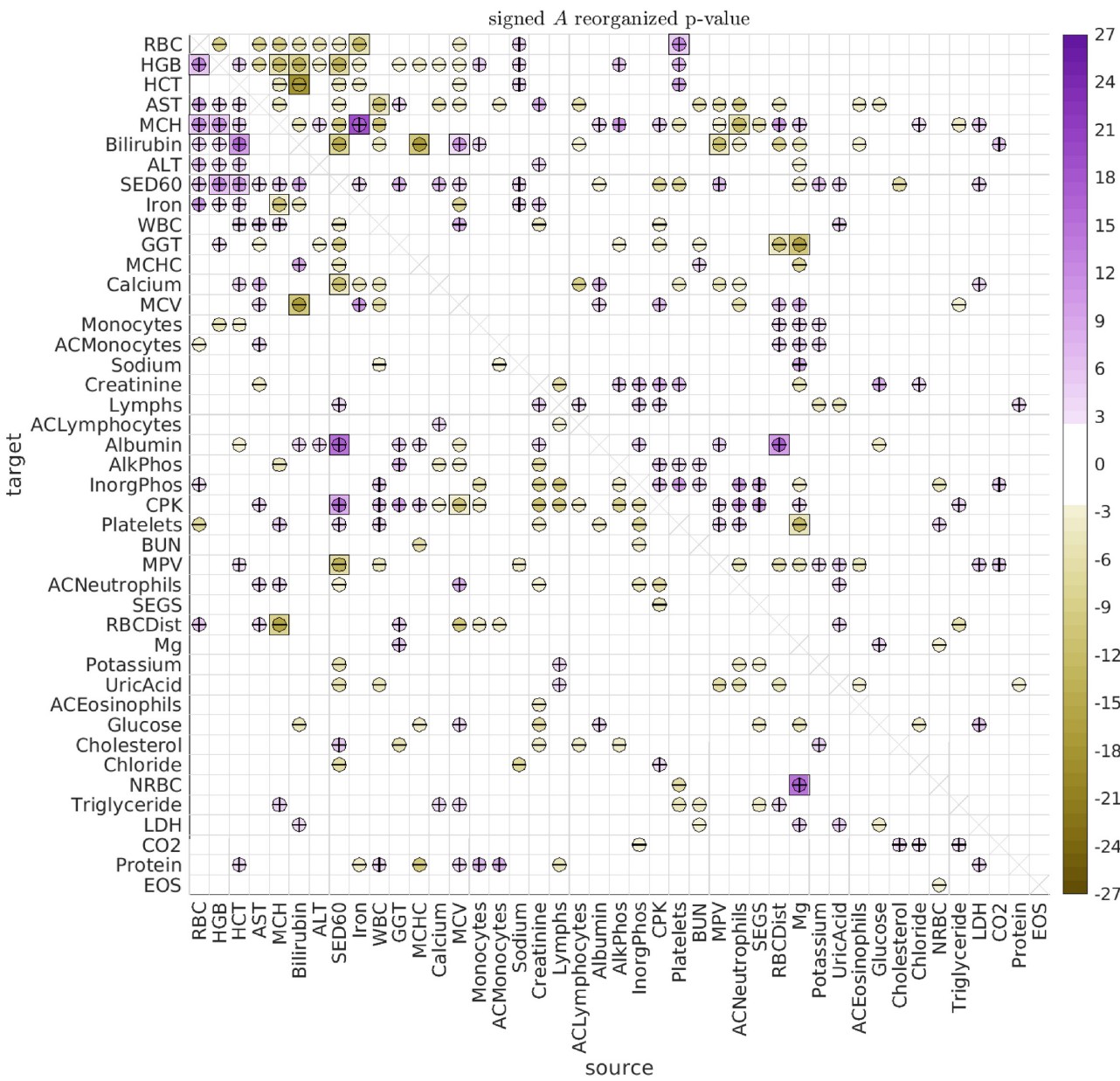

**Fig 8. Significant A rearranged.** This figure has the same format as Fig 6, except that we have rearranged the biomarkers to emphasize some of the most significant distinct push-pull blocks.

changes, such as the interaction between MCHC and RDW, and the interaction between Magnesium and Glucose.

The age-related changes we observe between RDW and MCV could reflect a change in the regulation of bone-marrow function, or the onset of certain kinds of iron-deficiency anemias [65–68]. More generally, RDW plays a role in various cardiovascular diseases, and it is possible that RDW serves as a proxy for the body's ability (or lack thereof) to tightly regulate certain processes [69–71]. Moreover, the management of RDW seems to be important for aging, with at least one study reporting a larger effect-size in men [72, 73].

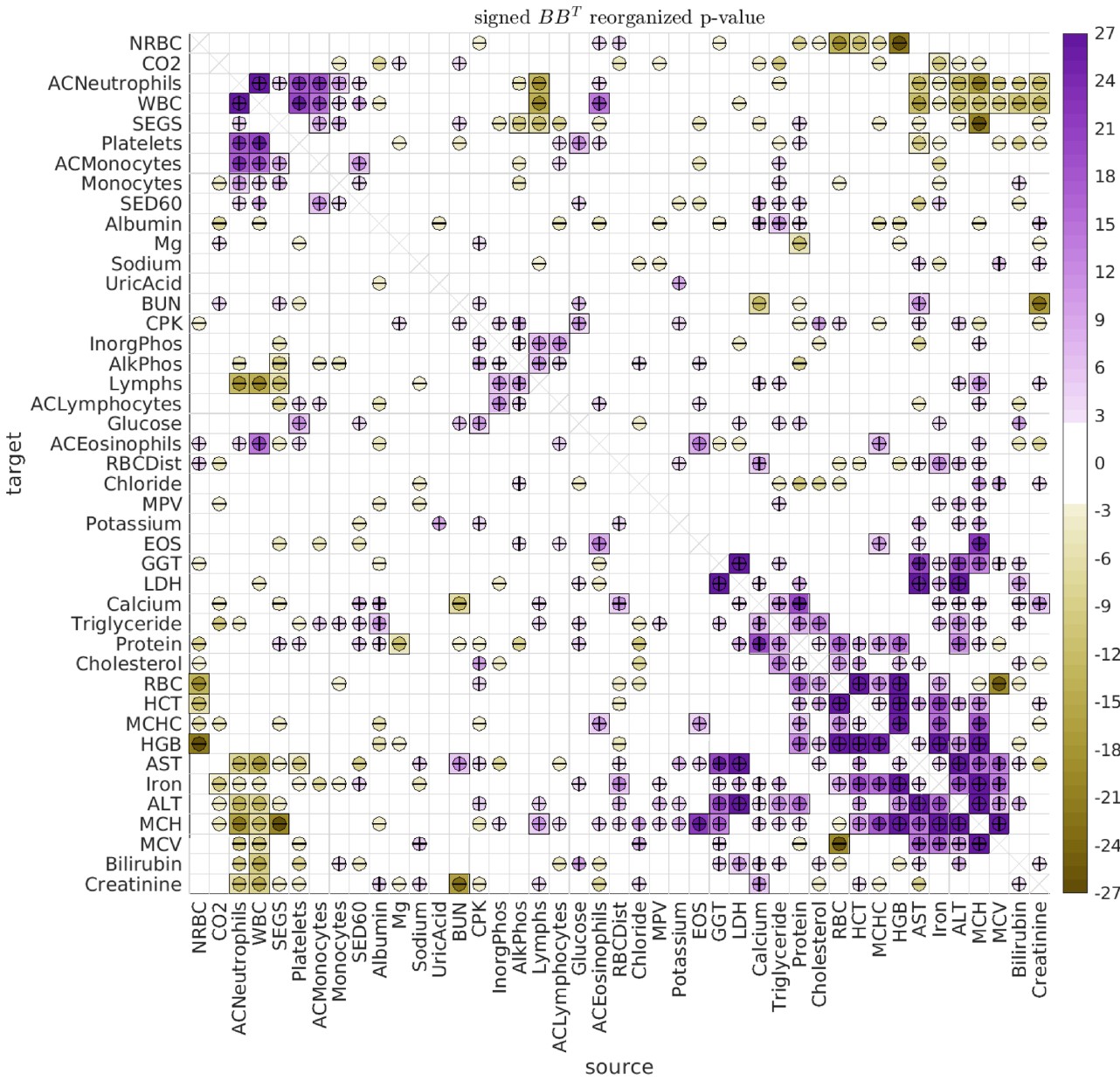

**Fig 9. Significant B rearranged.** This figure has the same format as Fig 7, except that we have rearranged the biomarkers to emphasize some of the most significant clusters.

Regarding the observed interaction between Magnesium and glucose, magnesium is known to affect blood glucose levels and glucose metabolism in humans, and was found to influence glucose absorption in rats [74–76]. Moreover, magnesium can play the role of a secondary messenger for insulin action, and magnesium accumulation is in turn influenced by insulin [77, 78]. Aging is also associated with changes in magnesium level, and magnesium deficits may result from many age-related conditions [79]. As just one example, the modulation of magnesium levels in elderly patients can potentially improve both insulin response and action, affecting blood glucose levels in turn [80].

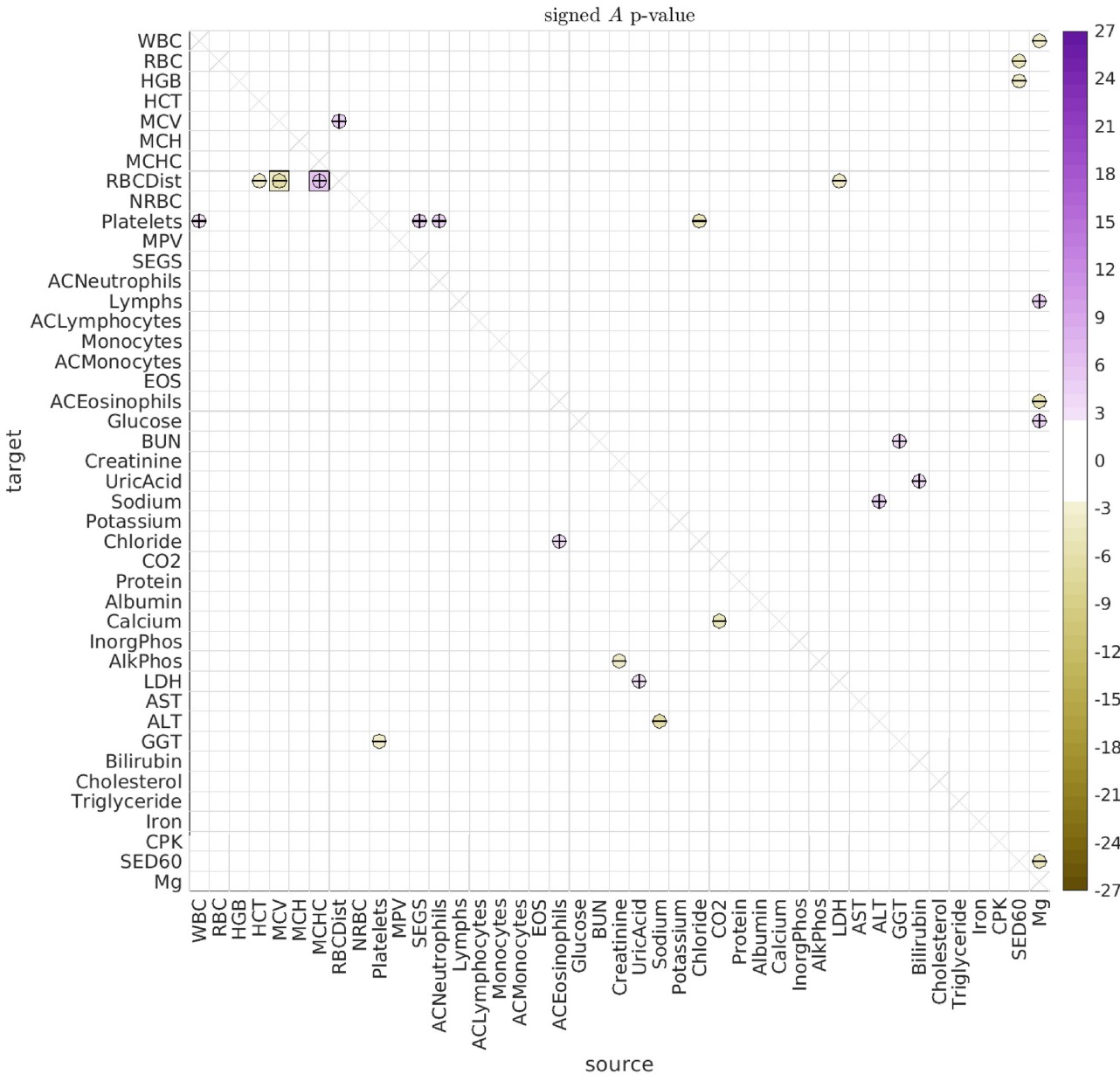

**Fig 10. Significant A for old vs young dolphins.** This figure illustrates the significant differences in the directed interactions $A_{vv'}$ between (i) dolphins over age 30 and (ii) dolphins between the ages of 10 and 30. Positive signs indicate that the corresponding interaction is more positive in the first group (i.e., dolphins over 30) than in the second group (i.e., dolphins between 10 and 30). Negative signs indicate the reverse.

## Discussion

We have used a linear SDE to model the time-series data from a longitudinal cohort of dolphins. Importantly, this model accounts for stochastic correlations between biomarkers, without which we would not have been able to accurately estimate the directed interactions.

After performing our analysis, we observed many statistically significant interactions across the population as a whole. Some of the most significant interactions involve 'push-pull' biomarker-pairs, where one biomarker excites the other while being inhibited in return. A large block of push-pull interactions involves (i) RBC, HGB and HCT interacting with (ii) AST,

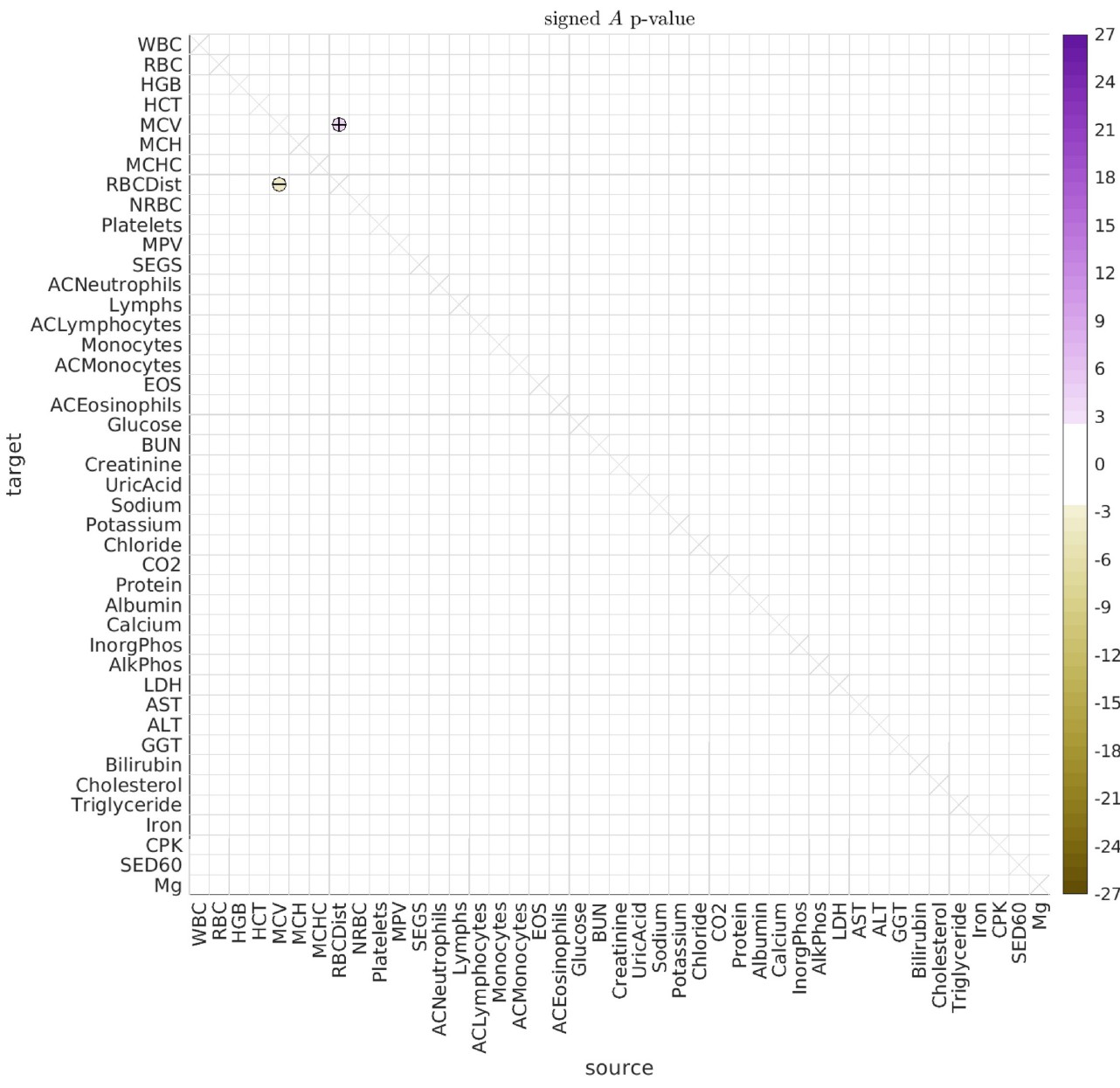

**Fig 11. Significant A for old male vs young male dolphins.** Here we illustrate significant differences in the directed interactions $A_{vv'}$ between (i) male dolphins over age 30 and (ii) male dolphins between the ages of 10 and 30.

MCH, Bilirubin, ALT, Sed60 and Iron, possibly relating to the relationship between liver function and RBC regulation.

We also reported several significant age-related changes to the interaction patterns. Some of the most significant age-related effects involve RDW, MCV and MCHC (in males and females), and the connection between magnesium and glucose (in females).

While certainly suggestive, our a-posteriori analysis cannot be used to decide whether these directed interactions are truly causal or not. Indeed, any significant type-A relationship between any pair of biomarkers (such as Iron and MCH shown in Fig 4) might be due to an unmeasured source affecting both biomarkers, but with different latencies. In this scenario

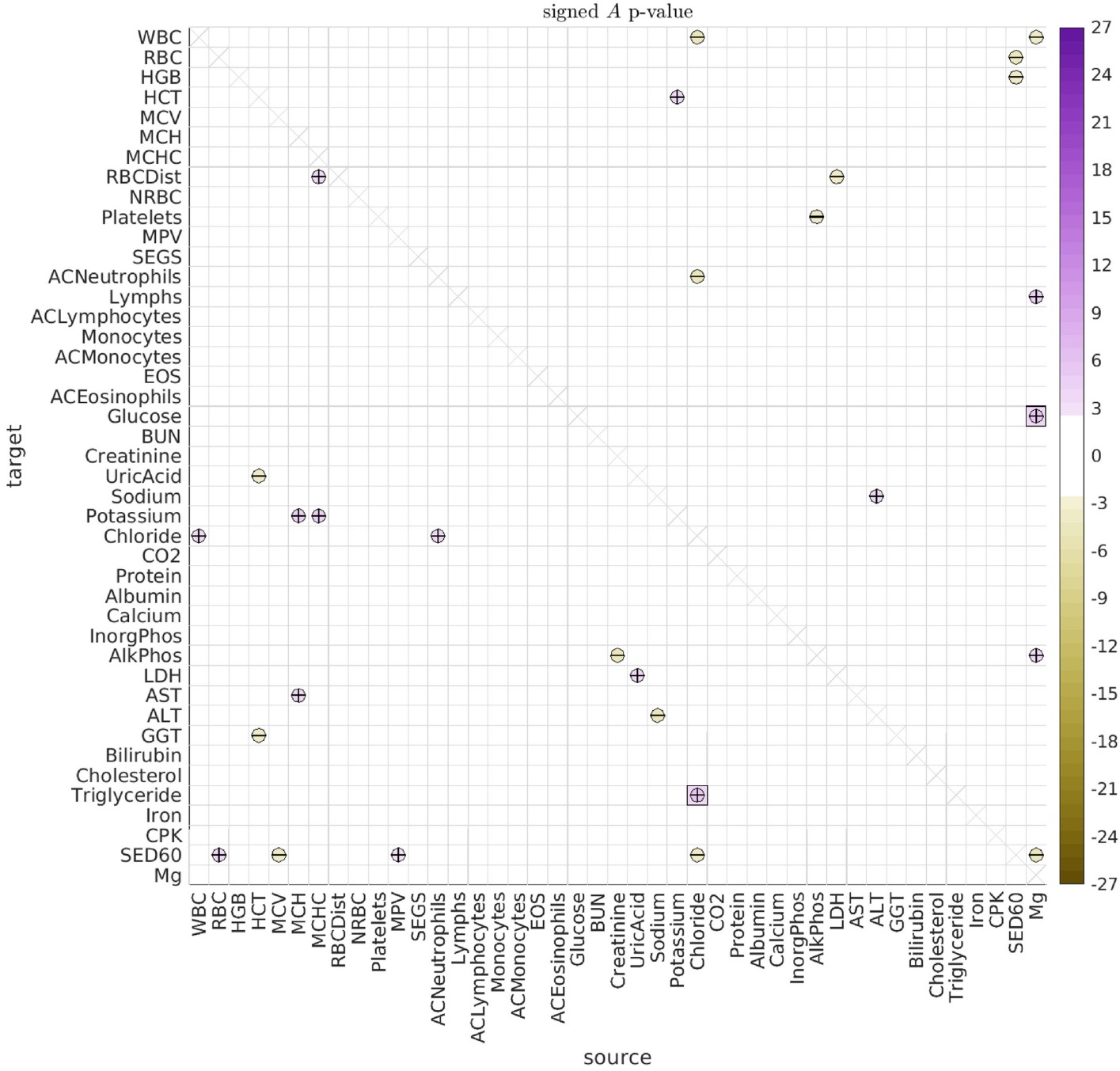

**Fig 12. Significant A for old female vs young female dolphins.** Here we illustrate significant differences in the directed interactions $A_{vv'}$ between (i) female dolphins over age 30 and (ii) female dolphins between the ages of 10 and 30.

both biomarkers would receive the same unmeasured signal, and would not be causally linked. However, the receiver with the shorter latency would appear to excite the receiver with the longer latency. Moreover, if the unmeasured signal was roughly oscillatory (e.g., meandering back and forth on a time-scale comparable to the difference in latencies), then the receiver with the longer latency would appear to inhibit the receiver with the shorter latency.

In summary, while we believe that most causal interactions would result in a signal captured by our SDE model, the reverse is certainly not true. Causality can only truly be determined after following up our analysis with a controlled study wherein one of the biomarkers is perturbed directly while measuring the response of the other biomarkers.

Finally, we remark that, while dolphins and humans share many age-related qualities [18], they are also different in important ways. For example, female dolphins do not experience menopause, and thus would not be expected to exhibit the suite of hormonal changes that human women do as they age [81]. Further work is needed to determine which of the interactions we have discovered are truly causal, which affect (or are affected by) aging, and which generalize to humans.

## Methods and materials

### Study population and animal welfare

This study was limited to archived, retrospective health data and biospecimen analyses that were collected on Navy bottlenose dolphins (*Tursiops truncatus*) as part of their routine health care. The US Navy Marine Mammal Program is an Association for Assessment and Accreditation of Laboratory Animal Care International-certified program that adheres to animal care and welfare requirements outlined by the Department of Defense and US Navy Bureau of Medicine.

### Sample and data collection

Archived data for this study were limited to measures obtained from routine blood samples collected from Navy bottlenose dolphins in the morning following an overnight fast between January 1994 and December 2018 ($n$ = 5889 samples from 144 dolphins). Blood samples collected either as an initial response or follow-up to acute clinical health concerns were excluded. Methods of routine blood sampling and the measurements obtained from the Navy dolphins have been described previously [18, 35, 64]. Data on the following 44 measures were available for analysis, of which we use $N$ = 43: red blood cell indices (RBC count (RBC), hemoglobin (HGB), hematocrit (HCT), mean corpuscular volume (MCV), mean corpuscular hemoglobin (MCH), mean corpuscular hemoglobin concentration (MCHC), RBC distribution width (RBCDist, or RDW), and nucleated RBCs (NRBC)); platelets and mean platelet volume (MPV); white blood cell count (WBC); eosinophils (EOS), lymphocytes (Lymphs), monocytes, and neutrophils (SEGS) (percent and absolute counts, with the latter prefixed by 'AC'); glucose, blood urea nitrogen (BUN), creatinine, uric acid, sodium, potassium, chloride, carbon dioxide ($CO_2$), total protein, albumin, calcium, inorganic phosphate (InorgPhos), alkaline phosphatase (AlkPhos), lactate dehydrogenase (LDH), aspartate aminotransferase (AST), alanine aminotransferase (ALT), gamma-glutamyl transpeptidase (GGT), bilirubin, total cholesterol, triglycerides, iron, creatine kinase (CPK), erythrocyte sedimentation rate (SED60), magnesium (Mg), and estimated glomerular filtration rate (GFR).

### Relationships between biomarkers

**Preprocessing.** For two of the biomarkers we excluded measurements with atypical lab-codes that lay outside the standard range of measurements seen from the two most common lab-codes. This involved removing Albumin measurements below 2, and RBC distribution width (RDW) measurements above 40. We also ignored the glomerular filtration rate (GFR) biomarker, as for this data-set the GFR measurements $v$ are functionally related to the

Creatinine measurements $v'$ (up to rounding/discretization error) as:

$$4 \log^2(v') = 3 \log^2\left(\frac{300}{v}\right). \tag{3}$$

Consequently, after applying the log-transformation described below, these two biomarkers become (almost) exactly correlated with one another.

**Log-transform.**   For each biomarker, we applied a log-transformation (adding the smallest discrete increment as a psuedocount) when such a transformation would reduce the skewness of the resulting distribution (across all dolphins). For this data-set the list of biomarkers undergoing a log-transformation is: WBC, MCV, RDW, NRBC, ACNeutrophils, Lymphs, ACLymphocytes, Monocytes, EOS, ACEosinophils, Glucose, BUN, Creatinine, UricAcid, Potassium, Protein, Calcium, AlkPhos, LDH, AST, ALT, GGT, Bilirubin, Cholesterol, Triglyceride, Iron, CPK, SED60, GFR.

**Normalization.**   For each dolphin we estimated the age-related drift for each biomarker by using linear-regression on measurements taken from age 5 onwards, excluding outliers above the 99th percentile or below the 1st percentile. We record the slope of this age-related linear drift for each biomarker for each dolphin, referring to this value later on to categorize individual dolphins as slow- or accelerated-agers (see Fig H in S1 Text). After removing this linear drift term, we normalize each biomarker (for each dolphin) to have mean 0 and variance 1.

**Analysis.**   For any particular subset of dolphins (e.g., male dolphins between the ages of 10 and 30), we fit a linear stochastic-differential-equation (SDE) to each pair of biomarkers, pooling observations across the dolphins in the chosen subset. Each of these SDEs has 12 parameters in total, accounting for type-A directed interactions, type-B shared stochastic drive, and type-C observation-noise (see Eqs (1) and (2)). Briefly, we performed the fit by using a variation of expectation maximization to approximate the model parameters that were most likely to have produced the observed data. The details are described in S1 Text. Our a-posteriori grouping of the type-A directed interactions (shown in Fig 8) is described in S1 Text. The code for these methods can be found within the github repository 'https://github.com/adirangan/dir_PAD'.

**Null Hypothesis.**   To estimate significance for any subset of dolphins across any age-interval for any pair of biomarkers, we compare the estimated parameters of our SDE-model to the parameters obtained after randomly permuting the time-labels of the measurements. Each label-shuffled trial $k = 1, \ldots, K$ is associated with a permutation of the time-labels $\pi_k$ which randomly interchanges time-labels within each dolphin (but not across dolphins) within the given age-interval. These label-shuffled trials correspond to sampling from the null hypothesis that the true interaction coefficient $A_{vv'}$ for any pair of biomarkers is 0, while still maintaining many of the correlations between biomarkers. Note that the same set of permutations $\{\pi_1, \ldots, \pi_K\}$ is used for the label-shuffled trials $1, \ldots, K$ across all biomarker-pairs $v, v'$. Consequently, we can use the correlations between different parameters estimated from the same label-shuffled trial to adjust the p-values of any given parameter (see $p_h$ below).

**Estimating $p$-values.**   For each parameter we estimate the $p$-value '$p_0$' numerically using $K = 256$ label-shuffled trials. When the parameter value for the original data is more extreme than the analogous parameter from any of the $K$ label-shuffled trials, we estimate $p_0$ by first (i) estimating the $z$-score $z_0$ for that parameter using the Gaussian-distribution fit to the collection of $K$ label-shuffled parameter-values, and then (ii) estimating the $p$-value $p_0$ by applying a 2-sided test to the $z$-score $z_0$; i.e., $\log(p_0) = \text{erfcln}(|z_0|/\sqrt{2})$. To correct for multiple hypotheses we calculate two adjusted p-values '$p_b$' and '$p_h$'. The adjusted $p$-value $p_b$ is obtained using a

standard bonferroni-correction. Thus, $p_b = p_0/J$, where $J$ is the number of parameters under consideration. For the directed interactions $A_{vv'}$ we set $J = N(N-1)$, while for the covariances $[BB^{\mathsf{T}}]_{vv'}$ we set $J = N(N-1)/2$ (recall $N = 43$ is the total number of biomarkers analyzed). This bonferroni-corrected $p$-value is an overestimate (i.e., $p_b$ is too conservative). A more accurate $p$-value can be obtained by using an empirical version of the holm-bonferroni adjustment, as described in S1 Text. These strategies can easily be extended to estimate the significance between different groups of dolphins (see Fig G in S1 Text).

**Estimating Aging-rate.**    To demonstrate consistency with the analysis of [35], we can estimate the aging rate of the dolphins. These results are shown in Fig H in S1 Text.

## Supporting information

**S1 Text. Description of the methods used in the main manuscript.**
(PDF)

**S1 Table. Signed logarithm of the bonferroni-corrected $p$-values for the A-type interactions shown in Fig 6 (comma-separated values).**
(CSV)

**S2 Table. Signed logarithm of the holm-bonferroni-corrected $p$-values for the A-type interactions shown in Fig 6 (comma-separated values).**
(CSV)

**S3 Table. Signed logarithm of the bonferroni-corrected $p$-values for the B-type interactions shown in Fig 7 (comma-separated values).**
(CSV)

**S4 Table. Signed logarithm of the holm-bonferroni-corrected $p$-values for the B-type interactions shown in Fig 7 (comma-separated values).**
(CSV)

**S5 Table. Signed logarithm of the bonferroni-corrected $p$-values for the A-type interactions shown in Fig 10 (comma-separated values).**
(CSV)

**S6 Table. Signed logarithm of the holm-bonferroni-corrected $p$-values for the A-type interactions shown in Fig 10 (comma-separated values).**
(CSV)

**S7 Table. Signed logarithm of the bonferroni-corrected $p$-values for the B-type interactions associated with Fig 10 (comma-separated values).**
(CSV)

**S8 Table. Signed logarithm of the holm-bonferroni-corrected $p$-values for the B-type interactions associated with Fig 10 (comma-separated values).**
(CSV)

**S9 Table. Signed logarithm of the bonferroni-corrected $p$-values for the A-type interactions shown in Fig 11 (comma-separated values).**
(CSV)

**S10 Table. Signed logarithm of the holm-bonferroni-corrected $p$-values for the A-type interactions shown in Fig 11 (comma-separated values).**
(CSV)

**S11 Table. Signed logarithm of the bonferroni-corrected *p*-values for the A-type interactions shown in Fig 12 (comma-separated values).**
(CSV)

**S12 Table. Signed logarithm of the holm-bonferroni-corrected *p*-values for the A-type interactions shown in Fig 12 (comma-separated values).**
(CSV)

## Author Contributions

**Conceptualization:** Aaditya V. Rangan, Caroline C. McGrouther, Nivedita Bhadra, Stephanie Venn-Watson, Eric D. Jensen, Nicholas J. Schork.

**Data curation:** Aaditya V. Rangan, Caroline C. McGrouther, Stephanie Venn-Watson, Eric D. Jensen, Nicholas J. Schork.

**Formal analysis:** Aaditya V. Rangan, Caroline C. McGrouther, Nivedita Bhadra, Nicholas J. Schork.

**Funding acquisition:** Aaditya V. Rangan, Stephanie Venn-Watson, Eric D. Jensen, Nicholas J. Schork.

**Investigation:** Aaditya V. Rangan, Stephanie Venn-Watson, Eric D. Jensen, Nicholas J. Schork.

**Methodology:** Aaditya V. Rangan, Caroline C. McGrouther, Nivedita Bhadra, Stephanie Venn-Watson, Eric D. Jensen, Nicholas J. Schork.

**Project administration:** Aaditya V. Rangan, Stephanie Venn-Watson, Nicholas J. Schork.

**Resources:** Aaditya V. Rangan, Stephanie Venn-Watson, Eric D. Jensen, Nicholas J. Schork.

**Software:** Aaditya V. Rangan.

**Supervision:** Aaditya V. Rangan, Nicholas J. Schork.

**Validation:** Aaditya V. Rangan.

**Visualization:** Aaditya V. Rangan.

**Writing – original draft:** Aaditya V. Rangan, Caroline C. McGrouther, Nivedita Bhadra, Stephanie Venn-Watson, Eric D. Jensen, Nicholas J. Schork.

**Writing – review & editing:** Aaditya V. Rangan, Nicholas J. Schork.

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
