## [Decision Letter · Decision Letter 0]

22 Oct 2022

Dear Dr. Rangan,

Thank you very much for submitting your manuscript "A time-series analysis of blood-based biomarkers within a 25-year longitudinal dolphin cohort." for consideration at PLOS Computational Biology.

As with all papers reviewed by the journal, your manuscript was reviewed by members of the editorial board and by several independent reviewers. In light of the reviews (below this email), we would like to invite the resubmission of a significantly-revised version that takes into account the reviewers' comments.

In particular, Reviewer #3 requested data and code for the model to be provided.

We cannot make any decision about publication until we have seen the revised manuscript and your response to the reviewers' comments. Your revised manuscript is also likely to be sent to reviewers for further evaluation.

Sincerely,

Mark Alber, Ph.D.

Section Editor

PLOS Computational Biology

Mark Alber

Section Editor

PLOS Computational Biology

Reviewer's Responses to Questions

**Comments to the Authors:**

Reviewer #1: In this study by Rangan et al., a time-series analysis of blood-based biomarkers within a 25-year longitudinal dolphin cohort was performed to disentangle the deterministic interactions between biomarkers, correlated stochastic fluctuations, and observation noise. The authors conducted a pairwise analysis by fitting two time series to a stochastic differential equation (SDE) model. The pairwise interactions among 44 biomarkers were reported. While this is an interesting work, I have a few questions on the approach and interpretations.

1. My major concern is about the identifiability of the model. Given the large number of unknown parameters, was the model fitting be able to recover the unique solution? This is particularly relevant when the “type-B correlations typically dominate the type-A interactions”. It might be good to perform an analysis using simulated time series and systematically investigate the identifiability. For instance, how much noise can be tolerated by the inference? What levels of signal-to-noise ratio can guarantee robust estimate? What if the signals of type-A and type-B interactions are strongly imbalanced?

2. I am wondering if the authors considered the potential temporal delays of biomarkers. Is this delay relevant in this study?

3. The authors stressed that the correlational analysis cannot be used to decide whether deterministic interactions are truly causal or not. This is true as only pairwise analysis was performed. Have the authors considered partial correlation, which controls for additional variables?

4. It would be helpful to plot the sample collection times for a few dolphins to get a better idea of the time intervals between samples. How is this interval related to the choice of the dt in SEDs?

5. The authors explained the biological processes underneath several identified interactions. There are a large number of interactions identified in the analysis. Could the authors discuss how to use the identified interactions in aging studies? Is there a principled way to analyze these interactions? It would be great to know how to extract patterns from the large number of interactions beyond visual inspection.

Reviewer #2: This manuscript conducts a rigorous statistical analysis of a rich longitudinal dataset pertaining to dolphin aging. The method distinguishes how biomarkers influence one another with a time delay (matrix A, described as “deterministic interactions”), how biomarkers co-vary during the process by which their values diffuse during aging (matrix B, described as “stochastic drive”), and non-time-dependent variation including short-term biological fluctuation and measurement error (matrix C). The manuscript finds that A cannot be accurately inferred without properly accounting for B and C, which the authors go ahead and do. This seems highly plausible, and the methods are appropriate for the task. Both A and B are of significant biological interest, although biological implications are a little underdeveloped in the paper.

My biggest concern is not with the analysis but with spanning the communication gap to biologists studying aging. Speaking for myself, I had a hard time following the text and graphical explanations of the method, but have enough math background to simply read Equations 1 and 2. I am concerned about readers with less math background who are unable to do that. I do not think such biologists will be able to make sense of the terms “deterministic interactions” or “stochastic drive”. The term “drift” (3 lines below Eq. 1) should also be avoided altogether because of the unfortunate series of events by which a diffusion rather than a directional term has come to be known as “random genetic drift” in biology.

I suggest instead discussing matrix A in terms of an aging “cascade” or “pathway”, which captures the time-dependent nature of the terms, and merely hints at their more deterministic nature without confusing readers who aren’t used to partitioning into deterministic and stochastic terms for the same process. These alternative terms would align nicely with the trajectory of Figure 5 left. Matrix B would be more comprehensible as the correlations that occur within “stochastic aging” (already known in some special cases as “epigenetic drift”, following the analogy with random genetic drift, which as I mention above, refers to the diffusion term rather than the drift term). Optionally, I think matrix B might be directly analogous to the mutation variance-covariance matrix M of quantitative genetics. I think these simple changes in terminology could have an important impact on the reception of the paper by the biologists of aging who I hope will be its audience.

Relatedly, Figure 5 left shows movement so slow through phase space that a cycle takes longer than a typical dolphin lives. It is possible, but hard to tell from Figure 5 right, whether smaller but more rapid oscillations also occur within individual dolphins. In any case, the Figure 4 schematic paints a picture of the latter rather than of the former, but if it is the former that is taking place, then “cascade” or “pathway” are definitely better metaphors than the metaphor of a stabilizing feedback loop from matrix A (stabilizing variation originating from matrix B) that is shown. As it stands, the Figure 4 schematic shows this stabilization as being sufficient to contain the fluctuations from matrix B altogether, which is obviously not generally the case e.g. in Figures 1-2. I don’t think that Figure 4 will further the reader’s correct understanding.

My other major comment is to suggest some additional work, which I see as optional. (My reviewing philosophy is to review the work you did, not what I wish you had also done.) I think the biological impact of the paper would be greater if it included some kind of dimensionality reduction to help extract more meaning from the inferred matrices A and B. It’s just hard work to extract biological meaning out of Figures 7-11, restricted as they are to pairwise interactions. There must be lots of different ways to paint a lower dimensional picture. E.g. to illustrate A, could you simulate with a dampened version of B and then take principal components or factors, and show which biomarkers load to each, interpret each component/factor, and replot with fewer dimensions? As I said, all optional, but I feel like something vaguely in this direction will eventually be needed to complete the bridge to biology. For matrix B / Figure 8, it might even be more straightforward. Or what about a simple eigendecomposition of these matrices: do the eigenvectors associated with the leading eigenvalues correspond to anything biologically sensible? My sense is that the authors are competent statisticians, so please don’t take my exact suggestions too literally, but I hope to give the gist of the kind of thing I think would help.

Typos (line numbering would make it easier to list these and other minor comments):

- Abstract “obervation”

- Throughout: Gaussian and Brownian should be capitalized.

Signed,

Joanna Masel

Reviewer #3: The manuscript has many strengths. The dataset of dolphin biomarkers is very interesting and may indeed have relevance to studies of human biology. I also commend the authors for articulating the concept that bodies are dynamical interacting systems. The SDE model is simplistic to ease interpretation of complex results. The paper is well-written. The notion that age effects are relevant is intriguing.

The manuscript does not make much of a contribution in a computational sense. No data or code are provided, and it is unclear how calculations were operationalized. The paper would be improved by describing code and making this code available to the community.

Minor comments: Please italicize Latin species names.

**Have the authors made all data and (if applicable) computational code underlying the findings in their manuscript fully available?**

Reviewer #1: Yes

Reviewer #2: **No: **The Data Availability statement points to the previously published dataset. There is no release of code, nor of post-processed data (e.g. the numerical values of the matrix elements).

Reviewer #3: **No: **No raw data or code were provided nor did the authors mention these items were placed in appropriate repositories.

PLOS authors have the option to publish the peer review history of their article (what does this mean?). If published, this will include your full peer review and any attached files.

Reviewer #1: No

Reviewer #2: **Yes: **Joanna Masel

Reviewer #3: No
---

## [Decision Letter · Decision Letter 1]

23 Jan 2023

Dear Dr. Rangan,

We are pleased to inform you that your manuscript 'A time-series analysis of blood-based biomarkers within a 25-year longitudinal dolphin cohort.' has been provisionally accepted for publication in PLOS Computational Biology.

Best regards,

Mark Alber, Ph.D.

Section Editor

PLOS Computational Biology

Mark Alber

Section Editor

PLOS Computational Biology

Reviewer's Responses to Questions

**Comments to the Authors:**

Reviewer #1: All my questions have been addressed. I appreciate the authors' efforts to revise the manuscript, especially the new analysis to explore the interaction pattern.

Reviewer #2: The authors have both addressed my concerns and also productively responded to those things that were mere suggestions. Only one last point - the readme in the github is a placeholder, and would greatly benefit from being fleshed out into a brief description of which scripts do what.

Signed,

Joanna Masel

Reviewer #3: All my concerns have been addressed. Nice paper!

**Have the authors made all data and (if applicable) computational code underlying the findings in their manuscript fully available?**

Reviewer #1: None

Reviewer #2: Yes

Reviewer #3: Yes

PLOS authors have the option to publish the peer review history of their article (what does this mean?). If published, this will include your full peer review and any attached files.

Reviewer #1: No

Reviewer #2: **Yes: **Joanna Masel

Reviewer #3: No

---

## [Editor Report · Acceptance letter]

8 Feb 2023

PCOMPBIOL-D-22-01227R1 

A time-series analysis of blood-based biomarkers within a 25-year longitudinal dolphin cohort.

Dear Dr Rangan,

I am pleased to inform you that your manuscript has been formally accepted for publication in PLOS Computational Biology. Your manuscript is now with our production department and you will be notified of the publication date in due course.

With kind regards,

Zsofia Freund
